# Polyploid Induction and Identification of *Rosa roxburghii* f. *eseiosa*

**DOI:** 10.3390/plants12112194

**Published:** 2023-06-01

**Authors:** Huijing Wu, Lanlan Jiang, Jin’e Li, Min Lu, Huaming An

**Affiliations:** 1Agricultural College, Guizhou University, Guiyang 550025, China; m15519080463@163.com (H.W.); j17385313153@163.com (L.J.); 17323700864@163.com (J.L.); 2National Forestry and Grassland Administration Engineering Research Center for Rosa roxburghii, Guiyang 550025, China

**Keywords:** chromosome doubling, colchicine, flow cytometry, *Rosa roxburghii* f. *eseiosa* Ku, stomatal characteristics

## Abstract

*Rosa roxburghii* f. *eseiosa* Ku is a variety of *Rosa roxburghii*, with two known genotypes: Wuci 1 and Wuci 2. The lack of prickle on the peel of *R. roxburghii* f. *eseiosa* makes it easy to pick and process, but its fruit size is small. Therefore, we aim to induce polyploidy in order to obtain a larger fruit variety of *R. roxburghii* f. *eseiosa*. In this study, current-year stems of Wuci 1 and Wuci 2 were used as materials for polyploid induction, which was carried out through colchicine treatment coupled with tissue culture and rapid propagation technology. Impregnation and smearing methods were effectively used to produce polyploids. Using flow cytometry and a chromosome counting method, it was found that one autotetraploid of Wuci 1 (2*n* = 4*x* = 28) was obtained by the impregnation method before primary culture, with a variation rate of 1.11%. Meanwhile, seven Wuci 2 bud mutation tetraploids (2*n* = 4*x* = 28) were produced by smearing methods during the training seedling stage. When tissue-culture seedlings were treated with 20 mg/L colchicine for 15 days, the highest polyploidy rate was up to 60%. Morphological differences between different ploidys were observed. The side leaflet shape index, guard cell length, and stomatal length of the Wuci 1 tetraploid were significantly different from those of the Wuci 1 diploid. The terminal leaflet width, terminal leaflet shape index, side leaflet length, side leaflet width, guard cell length, guard cell width, stomatal length, and stomatal width of the Wuci 2 tetraploid were significantly different from those of the Wuci 2 diploid. Additionally, the leaf color of the Wuci 1 and Wuci 2 tetraploids changed from light to dark, with an initial decrease in chlorophyll content followed by an increase. In summary, this study established an effective method for inducing polyploids in *R. roxburghii* f. *eseiosa*, which could provide a foundation for the breeding and development of new genetic resources for *R. roxburghii* f. *eseiosa* and other *R. roxburghii* varieties in the future.

## 1. Introduction

*Rosa roxburghii* f. *eseiosa* Ku is a forma of *R. roxburghii*, as clarified in *Microphyllae* of Rosaceae [1]. It was first discovered in Wenchuan County, Sichuan Province in 1975. There are typically five petals, and the fruits are nearly round, about 2 cm in diameter [1]. Compared to the main variety, ‘Guinong 5’, the fruit is smaller and the yield is lower [2]. However, the absence of prickles in the peel makes it a valuable fresh edible germplasm resource for the further breeding of large fruit germplasms. Additionally, it is suitable for cultivation, which can help with the issue of *R. roxburghii* fruit being solely used for processing and only having a single main cultivar.

Our laboratory has acquired abundant *R. roxburghii* resources from all over the country [3,4]. Among them, a germplasm resource was collected from Qianxi City, Guizhou Province that shares morphological characteristics with *R. roxburghii* f. *eseiosa* and was named Wuci 1. Through scanning electron microscopy, different forms of epidermal hairs were observed on the surface of Wuci 1’s peel, with thicker epidermal hair cell walls, lower cytoplasmic content, and a small number of osmiophilic particles in the cells [5,6]. Wuci 1’s single fruit weight is only 14 g, which is much smaller than the main variety ‘Guinong 5’ (30 g). However, its vitamin C content is similar to that of ‘Guinong 5’ [2]. Furthermore, the total phenolic content in Wuci 1’s leaves [7] and fruits [2] is higher than that of ‘Guinong 5’, with gallic acid identified as a significant metabolite that correlates with the fruit’s antioxidant capacity [8]. Specific molecular markers have been developed for Wuci 1, enabling rapid identification of both the variety itself and its offspring and reducing the breeding cycle [9]. Recently, another *R. roxburghii* f. *eseiosa* germplasm resource was collected from Sichuan Province, named Wuci 2 due to its consistency with the other characteristics of *R. roxburghii* f. *eseiosa*, except for the red fruit branch (Figure 1). Wuci 2 has not yet been systematically studied.

Polyploid breeding involves doubling the chromosome number of samples to create new genome variants. According to Islam [10], polyploids are crucial for developing desirable characteristics such as stress and disease resistance, higher yields, and superior quality. Polyploid breeding has promising applications in various crops, including apples [11], watermelon [12], and strawberries [13], resulting in an improved crop performance. Different plant tissues and organs, such as seeds [14], stems [15], and leaves [16], can be subjected to polyploid treatment. For instance, Sha [17] induced the seeds of ‘Guinong 5’ with colchicine, leading to obtaining three mutant plants. Feng [18] induced polyploidy in the seeds and seedlings of *R. roxburghii*, resulting in only two tetraploid plants. In contrast, Wang [19] obtained a 24.5% induction rate by treating the shoot tips of sterile *R. roxburghii* seedlings with colchicine. These findings suggest that polyploid induction through tissue culture could offer a higher polyploidy rate of *R. roxburghii*.

The results of this study will help understand the changes in ploidy levels among different genotypes of *R. roxburghii* f. *eseiosa* and their response to polyploid induction. These findings will serve as a foundation for future breeding attempts to enhance the content and yield of bioactive substances in the fruits of *R. roxburghii* f. *eseiosa* by using polyploid induction. Additionally, these results can serve as an optimization plan for in vitro polyploidization of *R. roxburghii* and other related species, leading to the development of superior genotypes.

## 2. Results

### 2.1. Induction of Polyploidy by Impregnating Stem Segments of R. roxburghii f. eseiosa before Primary Culture

After being treated with colchicine for 14 days, the stem segments of the Wuci 1 explant began to germinate, while the axillary buds of the Wuci 2 explant began to germinate after 10 days. However, this was about a week later than the control. The leaves of the germinated buds turned severely yellow and gradually lost their green color from the lower to the upper end of the plant. Eventually, the leaves withered and perished. The results from Table 1 indicate that the mortality rate progressively increased with a rise in colchicine concentration. Furthermore, the mortality rate observed in Wuci 1 was greater than that in Wuci 2. The mean mortality rate of stem segments was 44.66%, 62.51%, and 73.40% at three different treatment times.

Due to the high number of plant deaths during the cultivation process, it is evident from Table 1 that only a small number of plants successfully rooted and underwent refinement for subsequent ploidy identification. After conducting the ploidy identification, it was discovered that a mutant plant of Wuci 1 was produced using 500 mg/L colchicine and dimethyl sulfoxide (DMSO (1%)) for 12 h, resulting in a variation rate of 1.11%.

### 2.2. Induction of Polyploidy by Impregnating Stem Segments of R. roxburghii f. eseiosa before Subculture

Before culturing subcultures, it is necessary to impregnate the stem segments of tissue-cultured seedlings with buds. The results are displayed in Figure 1. As the concentration of colchicine treatment increases, the mortality rate of sterile stem segments of both Wuci 1 and Wuci 2 gradually rises. The mortality rate of stem segments in sterile seedlings of Wuci 1 is higher than that of sterile seedlings of Wuci 2. With an increasing colchicine concentration and fixed impregnation time, the survival rate of *R. roxburghii* f. *eseiosa* stem segments gradually decreases, but the degree of reduction is insignificant. However, when treatment concentration is constant, the mortality rate significantly increases with an extension of impregnation time. This suggests that compared to the concentration of colchicine treatment, the impregnation time causes greater damage to the stem segments of *R. roxburghii* f. *eseiosa*. The impregnation time of sterile seedling stem segments is longer than that of explant stem segments. The average mortality rates under the three treatment times are 44.66%, 62.51%, and 73.40%, respectively. After impregnating the stem segments of sterile seedlings with colchicine before subculture, the axillary buds that sprouted from the stem segments were smaller. However, during the subsequent cultivation process, a large number of plants died. Therefore, no mutant plants were obtained through this method.

### 2.3. Effect of Smearing Method on Tetraploid Induction of Wuci 2

According to Table 2, we selected Wuci 2 tissue culture seedlings that had been growing for three months and subjected them to treatment with 10–30 mg/L colchicine for 10–15 days. As a result, we obtained seven Wuci 2 mutant plants. The concentration range and treatment time used in the experiment did not affect the survival rate, but they did have a significant influence on the polyploidy rate. The optimal treatment condition was 20 mg/L of colchicine for 15 days, which resulted in a variation rate of 60%. Our observations of the treated plants revealed that those treated with 30 mg/L were able to produce new buds and true leaves initially, but the leaves progressively turned yellow and dried out over time. Axillary buds grew between the axils of the middle and lower leaves and grew into normal new shoots. After treatment, certain plants may undergo slow or abnormal growth, potentially attributed to colchicine’s inhibition of spindle formation and subsequent abnormal mitosis within the plant. Chromosome doubling may also upset the natural balance of diploid plants, leading to an imbalance of endogenous hormones and anomalous physiological and biochemical processes.

### 2.4. Flow Cytometry and Chromosome Counting Analysis of Mutant Plants

Flow cytometry was utilized to measure and analyze DNA content in plants exhibiting noticeable morphological variations. In Figure 2, the ordinate value represents the relative value of the measured cell number, the abscissa represents the fluorescence channel value, and the peak channel position reflects the ploidy level of the sample being evaluated. The diploid strain Wuci 1 exhibited a peak at channel 12,777.05, while the tetraploid strain Wuci 1 produced a peak at channel 23,123.87. This indicates that the tetraploid Wuci 1 had approximately twice the fluorescence intensity peak of diploid Wuci 1. Furthermore, the peak value of the analyzed plant map was concentrated without impurity, indicating that the suspected Wuci 1 polyploid tissue culture seedlings were tetraploid. By treating with 500 mg/L colchicine and adding DMSO (1%) for 12 h, a tetraploid Wuci 1 was obtained. The fluorescence intensity peak of the diploid Wuci 2 control was located at channel 13,026.17 (Figure 2A). In contrast, the peak fluorescence intensity of the seven tetraploid Wuci 2 samples was observed around channel 26,000 (Figure 2B). These peak intensities were approximately twice as high as those of the diploid Wuci 2 control.

By observing the root tip of diploid Wuci 1 and the shoot tip of Wuci 2, the chromosome numbers were determined. The chromosome number of diploid *R. roxburghii* f. *eseiosa* was 2*n* = 4*x* = 28 (Figure 2C). After the root tip observation, a homotetraploid Wuci 1 was observed, with a chromosome number of 2*n* = 4*x* = 28. After examining the shoot tip, tetraploid bud mutation Wuci 2 was detected, with a similar chromosome number of 2*n* = 4*x* = 28 (Figure 2D).

### 2.5. Morphological Comparison between Diploid and Tetraploid of R. roxburghii f. eseiosa

After reviewing Table 3, it is evident that the leaves of tetraploid *R. roxburghii* f. *eseiosa* induced by the soaking method exhibit no significant differences compared to diploids, with the exception of the significant difference in the leaf shape index of the side leaflets. The color of the diploid Wuci 1’s leaves are green, whereas the tetraploid Wuci 1’s leaf color change during growth and development. After one month, the plantlet’s leaf color becomes light green (Figure 3A1) with a relative chlorophyll content of 27.73 SPAD, which is significantly lower than that of the diploid Wuci 1. After two months, the leaf color becomes dark green (Figure 3A2) with a relative chlorophyll content of 40.21 SPAD, significantly higher than that of the diploid Wuci 1. Compared to diploid Wuci 1 stoma, tetraploid Wuci 1 stoma displays significant differences in guard cell length and stoma width. However, other indicators show no significant differences, as illustrated in Figure 4A,B. In the same cultivation conditions, the growth of the diploid Wuci 1 was better than that of the tetraploid Wuci 1, which showed growth lag. During subsequent growth, some mutagenic materials grew slowly or stopped growing.

In Figure 4, it is evident that the leaves of diploid Wuci 2 are long, elliptical, and green in color. On the other hand, tetraploid Wuci 2 exhibits two types of leaf shape variations primarily characterized by differences in shape and color, while the other phenotypic differences between diploid and tetraploid are negligible. The type I leaves show deformation, with the top edge of the leaves being deeply cracked, and the leaves take longer to develop. The type II leaves are elliptical in shape and are light green in color (Figure 3H1), with a later return to normal color (Figure 3H2). After reviewing Table 4, it is clear that a notable discrepancy exists between diploid Wuci 2 and tetraploid Wuci 2 in terms of the terminal leaflet shape index. Additionally, the relative chlorophyll content of the tetraploid demonstrates significant improvement compared to the diploid variety, increasing from 37.57 to 43.74 SPAD. After treating tetraploid Wuci 2 with 20 mg/L and 30 mg/L, it exhibited light leaf color, equivalent to tetraploid Wuci 1. Initially, the leaves turned yellow, and the relative chlorophyll content remained at 28.46 SPAD, significantly lower than that of diploid Wuci 2, but later returned to bright green with a relative chlorophyll content of 41.91 SPAD, displaying tetraploid characteristics. Compared to diploid Wuci 2, tetraploid Wuci 2’s guard cell length, guard cell width, stomatal length, and stomatal width increased by 21.21%, 14.17%, 24.25%, and 45.03%, respectively (Figure 4C,D). Tetraploid Wuci 2 has the characteristics of large stomata and low stomatal density typical of tetraploids, with diploid Wuci 2 showing a 1.34-times greater stomatal density than the tetraploid.

## 3. Discussion

### 3.1. Mutagenic Effects of Different Mutagenesis Methods on R. roxburghii f. eseiosa

The most commonly used mutagen in plants is colchicine [20,21]. Its mechanism of action mainly involves the regulation of microtubule stability. A microtubule is a cytoskeleton protein structure, which is involved in cell division, cell transport and cytoskeleton formation. The microtubule is made up of two different units, which are α-tubulin and β-tubulin. Colchicine blocks the polymerization of microtubules by binding and inhibiting tubulin dimer-α-β tubulin, making it unable to continue to form a stable cytoskeleton [22]. The failure of microtubule formation during mitosis will stop the cell cycle at the metaphase. This will prevent chromosome pairs from separating and moving to opposite poles during the anaphase to form two sets of chromosomes before cytokinesis. The failure of cytokinesis by the end of the cell cycle will cause two sets of chromosomes to remain in a single nucleus when the polyploid cell is formed [20,22,23,24,25]. Successful methods of chemical mutagenesis for inducing polyploidy include the impregnation method [26], mixed culture method [27], and smearing method [28], among others. Typically, young and well-differentiated parts such as clustered buds and callus tissues are used as induction materials [22]. In this study, the impregnation method caused variations in Wuci 1, while no variation was observed in Wuci 2, indicating that effective induction methods for polyploids of different genotypes within the same species vary. After three months of growth outside the tissue culture bottle, the smearing method caused variations in Wuci 2. 

Previous studies have successfully induced polyploidy in fruit trees, such as apple [29], jujube [30], and orange [31], using the mixed culture method. Therefore, the author believes in the potential of this method. The primary-culture mixed culture method and the subculture mixed culture method were also carried out, but no polyploidy was obtained, so the data were not included in this article. The survival rate of the impregnation method was higher than that of the mixed culture method, possibly because even though the colchicine concentration used in the mixed culture method was lower, the treatment time was longer. Thus, a large accumulation of colchicine in the plant may have occurred, which is consistent with Li’s [32] research findings. This study only used the impregnation method before the primary culture to obtain a homopolyploid plant of Wuci 1. The low polyploidy rate could be attributed to the weak vitality of mutagenized plants during tissue culture. Few may survive after rooting and hardening, and most may die in the bottle. Therefore, our polyploid identification was only performed after the rooting culture, resulting in a low variation rate.

In this study, both the mixed culture method and the impregnating method failed to induce polyploidy in Wuci 2. As a result, the author investigated and implemented a new method, the smear method, during the refining seedling stage of Wuci 2. The results were impressive, as the use of this method on only 30 seedlings yielded 7 mutant tetraploids, resulting in a polyploidy rate of 23%. When treated with 20 mg/L colchicine for 15 days, the variation rate peaked at 60%. This method has also produced high polyploidy rates in other plants such as *Brassica nigra* [33] and *Eucommia ulmoides* [34], with polyploidy rates of 73.33% and 63.3%, respectively. In contrast, previous studies on polyploid induction of *R. roxburghii* mostly utilized seeds as their experimental materials, resulting in a variation rate of only 0.09%, yielding two diploid and tetraploid chimeric plants and one diploid and octaploid chimeric plant [17]. Feng [18] treated germinated *R. roxburghii* seeds with 0.5% colchicine and 2% DMSO for 6 h and obtained two tetraploid plants with a variation rate of 2.5%. The highest variation rate observed was 12.9% in the cotyledons of roses treated with the smear method [35]. In summary, the impregnating seed method produced a low polyploidy rate in *R. roxburghii*, while previous studies have demonstrated that the smear method can achieve high polyploidy rates in various plants. Therefore, this study used the smear method and found it to be suitable for polyploid induction research on *R. roxburghii* f. *eseiosa*. Additionally, the mutation rate using the smear method was higher than the polyploidy rate found with the *R. roxburghii* seed impregnating method.

Among the three methods utilized in this study, the smear method displayed the highest induction rate, with each treatment exhibiting variations. The impregnating method, on the other hand, was simple and allowed for an ease of control regarding the experimental conditions. Additionally, the plants obtained through this method were homogeneous tetraploids, thereby reducing the need for purification processes for mixoploids. In contrast, the mixed culture method required a significant amount of colchicine, which is expensive and resulted in weak induced seedlings, thereby hampering subsequent experiments. The results established that the concentration of colchicine and the treatment duration had an impact on the polyploidy rate of *R. roxburghii* f. *eseiosa*. In general, a higher concentration and longer duration of treatment led to a higher polyploidy rate. However, exceeding a threshold resulted in increased mortality rates. Thus, determining the appropriate explants, treatment concentration, and duration for each plant species is critical in attaining viable polyploids [22]. The high polyploidy rate of the smear method indicates the significant potential for its widespread application. However, it is important to note that the tetraploid produced by this method remains attached to the original diploid plant and is, consequently, akin to a diploid-tetraploid chimera. This necessitates their separation through tissue culture. Furthermore, it is crucial to remain vigilant regarding reverse mutations in mutagenized plants. To obtain ploidy-stable materials from plants exhibiting reverse mutations, the stem segments from the mutagenized site must be inserted into the growth medium. Additionally, the stem tips generated through proliferation culture should be continuously trimmed. Subsequently, the tetraploid segment of the plant must be removed for further culture, and the germinated lateral buds must be identified as homozygous polyploids through flow cytometry.

### 3.2. The Identification of Polyploidy

Polyploid identification is an important step in polyploid induction since polyploids differ from diploids in morphology, anatomy, and physiology. In this study, plant morphology, leaf stomata, and chromosome number were used to identify polyploids. Flow cytometry is a simple and effective tool for ploidy analysis, capable of quickly and accurately identifying the ploidy of mutant plants without limitations on tissue or cell stage. Plant tissue such as leaves, stems, roots, flowers, peels, and seeds can be used for flow cytometry identification, and the required samples are minimal. Feng et al. [18] used flow cytometry to determine the ploidy level of polyploid adventitious buds induced by *Rosa multiflora* Thunb. var. *inermis*. However, results obtained by flow cytometry are not necessarily accurate, and chromosome counting is currently recognized as the most precise method of ploidy identification. Chromosome preparation is a complex operation, and only cells in the middle stage of division can be observed by microscope, causing difficulty in identifying polyploids earlier [36]. Thus, in this experiment, the morphological observation method was initially used to screen for mutant plants, followed by flow cytometry analysis. Finally, chromosome number identification was carried out to determine ploidy, greatly reducing the workload and ensuring result accuracy.

It takes several years for woody plants to grow after induction to obtain polyploid results in flowers and fruits. Therefore, identification of polyploids can only be performed at the initial stage through the observation of the stems and leaves, from which doubling and undoubling plants can be preliminarily distinguished. The leaves of polyploid seedlings typically exhibit characteristics such as being thick and large, having a reduced surface area, a darker color, and large stomata with reduced stomatal density. However, the thickness of stems and internodes are generally not noticeable. Thus, the observation and identification of polyploid seedling leaves are an effective method of polyploid preliminary identification. This study found that the leaf shape of diploid Wuci 1 and tetraploid Wuci 1 were similar, with only a slight increase in the side leaflet shape index observed in the tetraploid Wuci 1. On the other hand, the tetraploid and diploid leaves of Wuci 2 exhibit distinct leaf morphology, with some leaf surfaces wrinkled and some leaf colors changing, and significant differences between terminal and side leaflets. Two different genotypes of *R. roxburghii* f. *eseiosa* experience leaf color changes from deep to light and then to deep again. Additionally, unlike common polyploid induction characteristics, the relative chlorophyll content in the leaves of tetraploid plants changes from lower-than-diploid-plants to higher-than-diploid-plants as the plant grows. After polyploidization, the variance observed between Wuci 1 and Wuci 2 may be attributed to their respective genotypes. Studies have shown that the chlorophyll content of polyploid plants may increase or decrease depending on the changes of different genes during polyploidization and their effects on chlorophyll metabolic pathways. Furthermore, environmental factors such as light intensity, light quality, temperature, moisture, and other factors may affect the chlorophyll metabolism and content in polyploid plants. Thus, the phenomenon of the leaf color of *R. roxburghii* f. *eseiosa* changing from light to deep in this study warrants further research. Additionally, the leaf morphology of different tetraploid *R. roxburghii* f. *eseiosa* strains tends to be consistent in subsequent growth.

The size and density of stomata have a correlation with chromosome ploidy. Generally, the stomata of tetraploids are larger compared to those of diploids, and the density of stomata is lower than that of diploids. Therefore, the ploidy of a plant can be preliminarily judged based on the stomatal index [37,38]. Studies have previously shown that stomata of tetraploid figs were significantly larger than those of diploid figs, and the stomatal density was significantly lower than that of diploid figs [39]. Furthermore, the stomatal length and width of tetraploid *Lycium ruthenicum* increased, but the stomatal density decreased [40]. The results of this study were consistent with these findings, as the polyploids induced in this study had larger stomata and smaller stomatal density. In conclusion, the observation of stomatal characteristics can be an effective method of identifying polyploids. 

## 4. Materials and Methods

### 4.1. Plant Material and Tissue Culture

The experimental materials used in this study were stem segments of Wuci 1 and Wuci 2 that were grown in the *R. roxburghii* germplasm repository of Guizhou University, Guiyang, China (26°42.4080 N, 106°67.3530 E) (Figure 5). This study induced polyploidy in sterile seedlings of Wuci 1 and Wuci 2, which were cultivated in MS medium with 30 g/L sucrose and 6 g/L agarose at a pH of 5.8. The primary culture medium for Wuci 1 was MS + 2.5 mg/L 6-Benzylaminopurine (6-BA) + 0.1 mg/L naphthaleneacetic acid (NAA), while the proliferation medium used was MS + 1.5 mg/L 6-BA + 0.1 mg/L NAA. The rooting medium included 1/2 MS, Indole-3-butyric acid (IBA) 0.3 mg/L, and activated carbon 0.5 mg/L. For Wuci 2, the primary culture medium was MS + 1 mg/L 6-BA + 0.1 mg/L NAA, and the proliferation medium was MS + 0.20 mg/L 6-BA + 0.1 mg/L NAA. The rooting medium included 1/2 MS, IBA 0.1 mg/L, and activated carbon 0.5 mg/L. Following one month of acclimatization in the substrate (vermiculite: humus: perlite = 1:1:1), the seedlings were transferred to pots for cultivation [41].

### 4.2. Mutagenesis Methods

#### 4.2.1. Mutagenesis by the Impregnation Method

Referring to the method employed by Blasco [42], two impregnation techniques were utilized to induce polyploidy: impregnation before primary culture and impregnation before subculture; the specific steps are depicted in Figure 6. (1) After preliminary experiments, it was found that all the materials perished after the impregnating time surpassed 24 h. Therefore, in order to increase the mutation probability, the mutation time of the stem segment of the explant was shortened, and 1% DMSO was added to assist colchicine penetrate into the material in a shorter time. The impregnation method before primary culture involved the addition of 1% DMSO to the colchicine solution, and the subsequent impregnation of the stem segments with buds before primary culture. Colchicine concentrations of 300, 500, 800, 1000 and 1500 mg/L were employed, and treatment times of 6, 12 and 24 h were set. A completely randomized design was applied, comprising 15 treatment combinations of Wuci 1 and Wuci 2, respectively; all treatments were repeated three times. (2) The impregnation method before subculture entailed the impregnation of stem segments of sterile seedlings, upon reaching a height of approximately 5 cm, before subculture. Colchicine concentrations of 300, 500, 800, and 1000 mg/L were utilized, and treatment times of 24, 48 and 72 h were set. Similar to the impregnation method before primary culture, a completely randomized design was employed, consisting of 12 treatment combinations of Wuci 1 and Wuci 2, respectively; all treatments were repeated three times. The colchicine solution, after being sterilized and filtered, was placed on an ultra-clean workbench. Materials were then immersed in colchicine solutions of varying concentrations. After the requisite shaking time in the shaker, set at a rotating speed of 80 r/min and away from light, the impregnated stem segments were removed from the ultra-clean workbench and cleaned with sterile water, repeating the process four to six times, with each immersion lasting for three minutes. Following 28 days of inoculation, the number of fatalities for each combination was recorded, and the death rate was computed. After obtaining complete plants through the rapid propagation system, the next polyploid identification was conducted, the number of mutant plants was tallied, and the polyploidy rate was calculated.
Mortality rate = (number of mortality seedlings/number of inoculated seedlings) × 100%
Polyploidy rate = (variation number/survival number) × 100%

#### 4.2.2. Mutagenesis by Smearing Method

Referring to Wang’s [43] method, once the plant height of tissue culture seedlings of Wuci 2 reaches approximately 10 cm after hardening, a fully expanded leaf was carefully chosen. Lower and upper unexpanded leaves were then removed. The axillary bud growth point of the stem section was wrapped with cotton, and the injury connected to the stem segment was also wrapped with cotton. Then, it was covered with black plastic film. Using a 1 mL syringe, different concentrations of colchicine were injected into the cotton, as shown in Figure 7. Morning and evening, smear injections were conducted based on the cotton’s moisture level. Treatment concentrations were set at three gradients: 10, 20, and 30 mg/L, and treatment time was 10–15 days. After treatment, the plant was washed 5–6 times with distilled water and then transplanted into a new pot.

### 4.3. Polyploid Verification and Morphological Analysis

#### 4.3.1. Flow Cytometry Identification

The flow cytometry analysis was performed according to the methods of Marangelli [44] using the CyFlow Ploidy Analyser (Sysmex-Partec, Goerlitz, Germany) to analyze the leaves of *R. roxburghii* f. *eseiosa* polyploidy plants. Young leaf tissue samples of approximately 0.5 cm^2^ were extracted from each treated seedling, chopped with a blade in 200 μL of extraction buffer (CyStain^®^ UV Precise P, Sysmex-Partec, Goerlitz, Germany), and then filtered through a 30 µm mesh filter (CyStain^®^ UV Precise P, Sysmex-Partec, Goerlitz, Germany) to remove debris. Afterward, 800 μL of DAPI staining solution (CyStain^®^ UV Precise P, Sysmex-Partec, Goerlitz, Germany) was added. 

#### 4.3.2. Chromosome Number Identification

Following the methods of Lin [45] and Meng [46], we conducted confirmatory chromosome counts on flow-cytometry-identified tetraploids. We collected stem tips (5 mm long) and root tips (5 mm long) and then meticulously cleaned any impurities attached to the materials. We pre-treated the root tip with a 0.004 mol/L 8-hydroxyquinoline solution for 4–6 h, and the stem tip with a saturated para dichlorobenzene for 2–4 h. The pretreated root and shoot tips were then placed in a Carnoy’s fluid (ethanol: glacial acetic acid, 3:1, *v*:*v*) overnight. We dissociated the fixed root tips in a dissociation solution (alcohol: HCl (36~38%), 1:1, *v*:*v*) for 6 min and the stem tips in 1 mol/L HCl in a 60 °C water bath for 20 min. Next, we cleaned the materials with distilled water thrice, mounted them on a slide, and added 1–2 drops of carbol fuchsin. After staining for 20 min, we applied a conventional pressure method with filter paper, gently tapped the samples with a pen, and observed and photographed them using a Leica light microscope (ICC50W Wetzlar Germany).

#### 4.3.3. Morphological and Physiological Analysis

Referring to the method used by Feng et al. [18], various morphological traits, including terminal leaflet length, terminal leaflet width, side leaflet length, side leaflet width, and leaf thickness, were measured for both diploid and polyploid strains of *R. roxburghii* f. *eseiosa* at the same growth stage. The leaves were selected from both diploid and polyploid strains of *R. roxburghii* f. *eseiosa* and the middle portion of the leaves was used for stomatal detection. The number of five stomata on each leaf of the test plant was calculated using a 40-times objective lens, and the average value was recorded as the stoma density. Furthermore, the length and width of ten protective cells for each plant were measured, and the average value was taken. To measure the relative chlorophyll content of ten leaves, a chlorophyll meter SPAD-502 (Minolta, Osaka, Japan) was used. In addition, the length and width of the guard cell and stoma were measured by peeling the epidermis from the middle of the leaf along its length and placing it on a microscope slide in water. Finally, the length and width of guard cells and stomata were measured using an eyepiece micrometer. All the experiments were repeated three times.

### 4.4. Data Analysis

In this study, the data were analyzed using IBM SPSS Statistics 22.0. The data were assessed using ANOVA with the Tukey HSD test or the Pearson correlation analysis. Statistical differences were considered significant if *p* < 0.05. The results were expressed as mean ± standard deviation. All measurements were repeated three times. The tables and graphs were created using Microsoft Excel 2010 and Adobe Photoshop CC 2023.

## 5. Conclusions

In this study, polyploid induction methods for *R. roxburghii* f. *eseiosa* tissue culture seedlings were established, using the lethal rate and polyploid induction rate as indicators. A total of one Wuci 1 tetraploid and seven Wuci 2 tetraploids were obtained. The Wuci 1 polyploid induced by the impregnation method is a homogeneous tetraploid, while the Wuci 2 polyploid induced by the smear method is a bud mutation tetraploid. Compared to diploid, the typical characteristics of tetraploid *R. roxburghii* f. *eseiosa*, including Wuci 1 and Wuci 2, are a darker leaf color and a decrease in chlorophyll content followed by an increase. The induction method used in this study has significant implications for plant breeding and variety improvement. Our successful induction of tetraploid *R. roxburghii* f. *eseiosa* is an important step in cultivating large fruit varieties of *R. roxburghii*, which will benefit fruit farmers and consumers alike.

## Figures and Tables

**Figure 1 plants-12-02194-f001:**
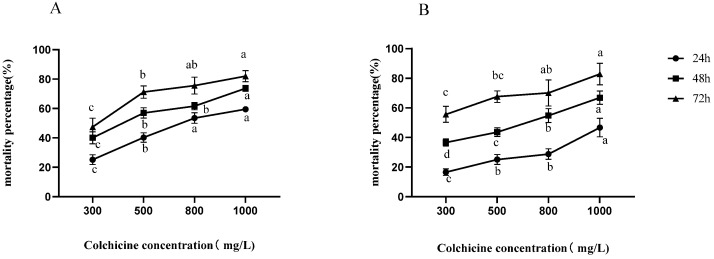
Effect of immersion before subculture on mortality rate of Wuci 1 (**A**) and Wuci 2 (**B**). Note: Three replicate experiments were performed; error bars represent standard error, means ± SD (*n* = 3). Different lowercase letters in the same curve indicate significant differences between the groups (*p <* 0.05).

**Figure 2 plants-12-02194-f002:**
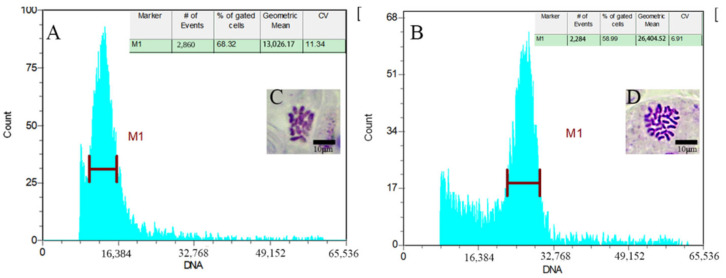
Flowcytometry analysis of *R. roxburghii* f. *eseiosa.* (**A**) Diploid and (**B**) tetraploid plant, depicting relative DNA content along with chromosomes under 100× magnification for (**C**) diploid and (**D**) tetraploid plant.

**Figure 3 plants-12-02194-f003:**
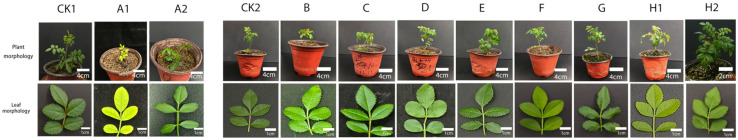
Comparison of plant morphology between diploid and tetraploid of *R. roxburghii* f. *eseiosa.* (**CK1**) Diploid Wuci 1; (**A1**,**A2**) Tetraploid Wuci 1; (**CK2**) Diploid Wuci 2; (**B**–**H2**) Tetraploid Wuci 2. Note: (**A1**) is the tetraploid Wuci 1 growing for 1 month in vitro, (**A2**) is the tetraploid Wuci 1 growing for 2 months in vitro. (**H1**) is the tetraploid Wuci 2 growing for 1 month in vitro, (**H2**) is the tetraploid Wuci 2 growing for 2 months in vitro.

**Figure 4 plants-12-02194-f004:**
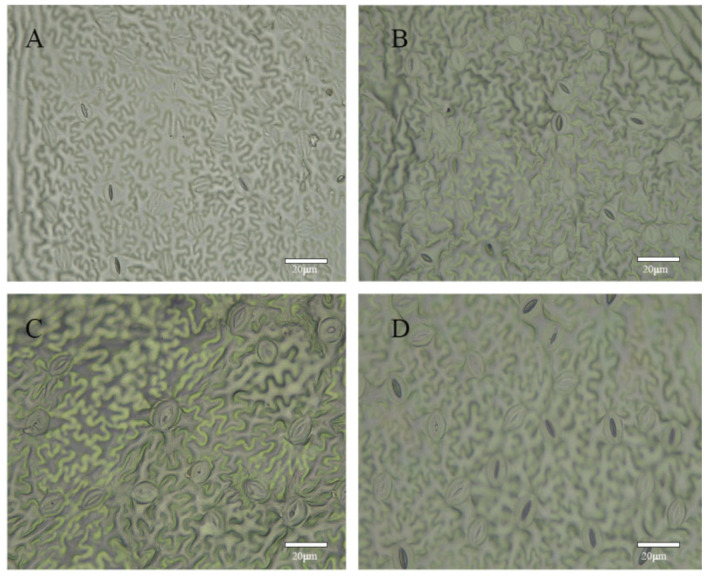
Stomatal characteristics of polyploid and diploid plants of *R. roxburghii* f. *eseiosa* (**A**) Stomata of diploid plant of Wuci 1; (**B**) stomata of tetraploid of Wuci 1; (**C**) stomata of diploid plant of Wuci 2; (**D**) stomata of tetraploid of Wuci 2.

**Figure 5 plants-12-02194-f005:**
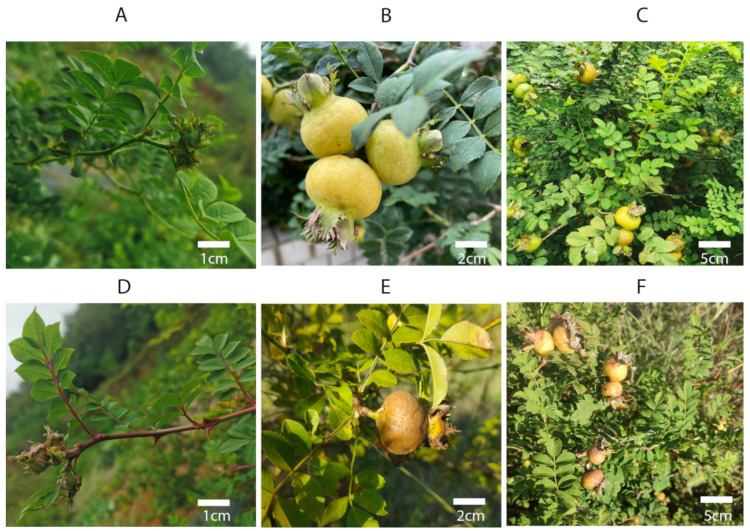
Stem and fruit morphology of Wuci 1 and Wuci 2. (**A**) Current year stem segment of Wuci 1; (**B**) fruits of Wuci 1; (**C**) plants of Wuci 1; (**D**) current year stem segment of Wuci 2; (**E**) fruits of Wuci 2; and (**F**) plants of Wuci 2.

**Figure 6 plants-12-02194-f006:**
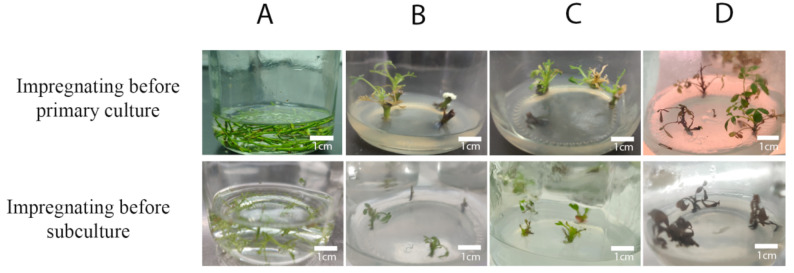
Treatment of *R. roxburghii* f. *eseiosa* by the impregnation method. (**A**) Colchicine impregnating material (**B**) Wuci 1 after 28 days of treatment; (**C**) Wuci 2 after 28 days of treatment; and (**D**) Wuci 1 after 28 days of subculture for the second time.

**Figure 7 plants-12-02194-f007:**
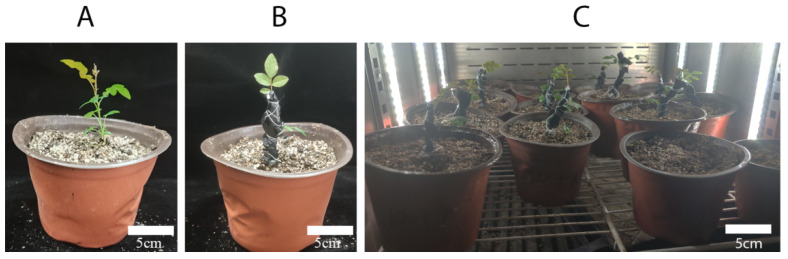
Smear treatment of Wuci 2. (**A**) Tissue culture seedlings of Wuci 2 grown for 3 months; (**B**) Wuci 2 after all removal except one leaf at the top; and (**C**) the plant growth state of Wuci 2 after Smear treatment.

**Table 1 plants-12-02194-t001:** Effect of impregnating before primary culture on stem-segment mortality of Wuci 1 and Wuci 2.

Genotype	Processing Time	Treatment Concentration (mg/L)	Vaccination Number	Mortality Rate (%)	Survival Number	Number of Survival Seedlings	Number of Variations	Polyploidy Rate (%)
Wuci 1	6 h	300	180	25.15 ± 3.35 c	135	18	0	0.00
		500	180	38.99 ± 4.22 b	109	10	0	0.00
		800	180	39.93 ± 2.71 b	108	8	0	0.00
		1000	180	43.16 ± 2.97 b	100	4	0	0.00
		1500	180	50.15 ± 3.35 a	90	4	0	0.00
	12 h	300	180	40.18 ± 3.21 d	104	14	0	0.00
		500	180	50.87 ± 2.39 c	90	15	1	1.11
		800	180	54.32 ± 3.80 bc	81	8	0	0.00
		1000	180	58.93 ± 3.09 b	73	1	0	0.00
		1500	180	75.48 ± 3.86 a	43	3	0	0.00
	24 h	300	180	69.07 ± 4.49 d	54	9	0	0.00
		500	180	73.51 ± 4.92 bc	46	7	0	0.00
		800	180	78.77 ± 3.59 ab	39	7	0	0.00
		1000	180	80.95 ± 5.45 ab	34	3	0	0.00
		1500	180	87.06 ± 2.48 a	21	2	0	0.00
Wuci 2	6 h	300	180	21.88 ± 3.13 d	140	21	0	0.00
		500	180	33.90 ± 6.15 c	118	17	0	0.00
		800	180	47.36 ± 2.32 b	93	10	0	0.00
		1000	180	51.11 ± 3.68 ab	86	9	0	0.00
		1500	180	57.09 ± 2.78 a	75	6	0	0.00
	12 h	300	180	42.85 ± 3.82 d	102	8	0	0.00
		500	180	58.22 ± 2.28 c	73	6	0	0.00
		800	180	61.76 ± 4.29 bc	68	6	0	0.00
		1000	180	68.97 ± 1.24 b	55	3	0	0.00
		1500	180	76.90 ± 5.13 a	41	1	0	0.00
	24 h	300	180	53.42 ± 3.58 d	82	6	0	0.00
		500	180	63.86 ± 1.20 c	64	3	0	0.00
		800	180	72.91 ± 1.86 c	41	3	0	0.00
		1000	180	78.19 ± 3.13 ab	37	2	0	0.00
		1500	180	82.27 ± 3.22 a	30	0	0	0.00

Note: Three replicate experiments were performed. Different lowercase letters in the mortality rate column indicate significant differences among the 5 treatment concentrations under 6, 12 and 24 h processing time (*p <* 0.05). The data presented are mean ± SD (*n* = 3).

**Table 2 plants-12-02194-t002:** Effect of smearing treatment on Wuci 2.

Colchicine Concentration (mg/L)	Processing Time/d	Number of Processes	Number of Survivors	Survival Rate (%)	Number of Variations	Polyploidy Rate (%)
0	10	5	5	100.00	0	0.00
	15	5	5	100.00	0	0.00
10	10	5	5	100.00	0	0.00
	15	5	5	100.00	1	20.00
20	10	5	5	100.00	1	20.00
	15	5	5	100.00	3	60.00
30	10	5	5	100.00	1	20.00
	15	5	5	100.00	1	20.00

**Table 3 plants-12-02194-t003:** Comparison of morphological characteristics between the diploid and tetraploid Wuci 1.

Indicators	Diploid	Tetraploid
Terminal leaflet length (mm)	18.81 ± 0.31 a	19.55 ± 0.49 a
Terminal leaflet width (mm)	10.98 ± 0.27 a	11.47 ± 0.25 a
Terminal leaflet shape index	1.72 ± 0.02 a	1.71 ± 0.02 a
Side leaflet length (mm)	18.62 ± 0.51 a	17.48 ± 0.41 a
Side leaflet width (mm)	10.06 ± 0.24 a	9.67 ± 0.29 a
Side leaflet shape index	1.71 ± 0.02 b	1.81 ± 0.04 a
Leaflet thickness (mm)	0.15 ± 0.01 a	0.14 ± 0.01 a
Relative content of chlorophyll (SPAD)	36.73 ± 0.64 b	27.73 ± 0.59 * c 40.21 ± 0.56 ** a
Length of guard cells (μm)	16.54 ± 0.22 b	18.25 ± 0.30 a
Width of guard cell (μm)	11.02 ± 0.36 a	11.54 ± 0.0.21 a
Stomatal length (μm)	10.90 ± 0.30 a	11.56 ± 0.37 a
Stomatal width (μm)	3.48 ± 0.13 b	3.933 ± 0.15 a
Stomatal density (number/10 × 4)	21.62 ± 0.58 a	19.67 ± 0.62 a

Note: * The tetraploid Wuci 1 growing for 1 month in vitro, ** the tetraploid Wuci 1 growing for 2 months in vitro. Three replicate experiments were performed. Different lowercase letters in the same row indicate significant differences between the diploid and tetraploid Wuci 1 (*p <* 0.05). The data presented are mean ± SD (*n* = 3).

**Table 4 plants-12-02194-t004:** Comparison of morphological characteristics between diploid and tetraploid Wuci 2.

Indicators	Diploid	Tetraploid
Terminal leaflet length (mm)	15.96 ± 0.20 a	16.46 ± 0.15 a
Terminal leaflet width (mm)	8.60 ± 0.17 b	9.54 ± 0.16 a
Terminal leaflet shape index	1.86 ± 0.04 a	1.74 ± 0.04 b
Side leaflet length (mm)	14.69 ± 0.22 a	14.90 ± 0.17 a
Side leaflet width (mm)	8.22 ± 0.75 b	8.98 ± 0.15 a
Side leaflet shape index	1.78 ± 0.05 a	1.66 ± 0.03 a
Leaflet thickness (mm)	0.14 ± 0.01 a	0.14 ± 0.01 a
Relative content of chlorophyll (SPAD)	37.57 ± 1.20 b	43.74 ± 0.98 a (28.46 ± 0.67 * c 41.91 ± 1.41 ** a)
Length of guard cells (μm)	17.30 ± 0.29 b	20.97 ± 0.35 a
Width of guard cell (μm)	11.86 ± 0.25 b	13.54 ± 0.20 a
Stomatal length (μm)	11.26 ± 0.41 b	13.93 ± 0.35 a
Stomatal width (μm)	3.22 ± 0.17 b	4.67 ± 0.12 a
Stomatal density (number/10 × 4)	22.76 ± 0.79 a	16.95 ± 0.55 b

Note: * Leaf yellowing plants of tetraploid Wuci 2 growing for 1 month in vitro, ** the chlorotic plants of tetraploid Wuci 2 grew green after 2 months in vitro. Three replicate experiments were performed. Different lowercase letters in the same row indicate significant differences between the diploid and tetraploid Wuci 2 (*p <* 0.05). The data presented are mean ± SD (*n* = 3).

## Data Availability

The data is contained within the manuscript.

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
