# Peer review of "Polyploid Induction and Identification of *Rosa roxburghii* f. *eseiosa"

_plants, 2023, doi:10.3390/plants12112194_

Round 1
Reviewer 1 Report
In this manuscript, Wu et al. describe the induction of polyploidy by colchicine treatment of stems of two diploid genotypes, Wuci 1 and Wuci 2, of Rosa roxburghii f. eseiosa plants. The authors test different experimental conditions (a range of colchicine concentrations and application times), combining two application methods (impregnation and smearing) with tissue culture and in vitro propagation techniques. They report the production of one autotetraploid from Wuci 1 and seven Wuci 2 bud mutation tetraploids, identified and characterised by determining several morphological parameters, flow cytometry and chromosome counting.
The generation of polyploids by colchicine treatment is a well-known procedure previously applied to many species, including R. roxburghii. Also, it is doubtful that the specific methods employed here can be directly applied to other taxa since, as rightly pointed out by the authors, the response is highly genotype-dependent. Therefore, the novelty and purely scientific interest of this work is very limited. Nevertheless, the generation of tetraploids in these particular genotypes could have some economic importance considering characteristics of commercial interest of polyploid crop plants, in general, and those of R. roxburghii, in particular, mentioned in the Introduction.
Unfortunately, the manuscript has several limitations and drawbacks and will require extensive modifications and rewriting before it can be considered for publication in Plants. The most relevant issues to be taken into account when preparing a revised version of the manuscript are detailed below (although the list is not exhaustive):
- To increase the readers' interest in the reported work, the authors should include in the Introduction (or maybe in the Discussion) a list of specific commercial applications of the obtained tetraploids and their possible use in breeding programmes. Some information on this issue is mentioned in the manuscript, but it should be highlighted more clearly.
- Please follow the journal instructions regarding the organisation and layout of the article. For example, Section 2 should be 'Results', not 'Material and Methods'.
- Define all abbreviations at first use in the text.
- In some cases, the description of the experimental procedures is unclear. For example, the authors indicate (section 2.3.1) that they measure several growth and morphological parameters in 'diploid and polyploid strains of R. roxburghii', which is confusing since these measurements are supposed to be used to identify the polyploids. Please, check carefully and rewrite the text where necessary.
- 'Chlorophyll meter' (line 173): indicate model, company, city and country.
- The statistical analyses are, in general, poorly described. In most cases, the number of biological replicas used in the experiments is unclear; it is not mentioned in section 2.4 (Data analysis), in the text or in the legends of figures and tables (n = ?). Also, it is nowhere indicated if the numbers in tables and the error bars in figures represent SE or SD.
- Tables and figures should be understood without reference to the text, including all relevant information in the table/figure itself or the corresponding legend. The legends of all tables and figures in the manuscript should be extended to include this information: definition of abbreviations and variables that are not self-evident, definition of all panels in a figure (e.g., panels CK1, CK2, and B to G in Fig. 5), statistical analysis (means ± SE or SD, n = ?), etc.
- Some specific comments:
Table 1: it is unclear; include horizontal lines to separate the different times of treatments and Wuci 1 from Wuci 2.
Fig. 3C is unclear. What do you mean by 'processing diagram'?
Fig. 4: Y-axis, 'mortality percentage'.
Fig. 5: The panels showing 'number of stomata', 'DNA content' and 'chromosome number' are too small, and it is not possible to observe the details of the photos. The figure should be modified, or these panels should be shown in independent figures, in any case of larger size.
- Do not repeat in the text the numerical values of measured parameters that are included in the tables (e.g., in sections 3.1 or 3.4) since this is redundant and unnecessary.
- The information in the Discussion is interesting, but this section is too long; the text should be shortened.
- Please, use a uniform format for the list of references, following the journal style. For example, the names of the journals are sometimes written in lowercase font and sometimes in capital letters.
The English language is generally correct. No significant grammar or spelling mistakes have been detected. Nevertheless, the text should be checked to try and improve some sentences that may be somewhat unclear.
Author Response
Dear editor,
Thank you for the positive and constructive comments on our paper entitled “Polyploid induction and identification of Rosa roxburghii f. eseiosa”, Manuscript ID: plants-2393344.
I am very grateful to the reviewers for their suggestions. We have substantially revised our manuscript based on the comments provided by the two reviewers. We have prepared a revised manuscript file with tracked changes and a clean manuscript file. Our responses to the reviewers are outlined below:
Responses to the comments of Reviewer #1
- In this manuscript, Wu et al. describe the induction of polyploidy by colchicinetreatment of stems of two diploid genotypes, Wuci 1 and Wuci 2, of Rosa roxburghii f. eseiosa plants. The authors test different experimental conditions (a range of colchicine concentrations and application times), combining two application methods (impregnation and smearing) with tissue culture and in vitro propagation techniques. They report the production of one autotetraploid from Wuci 1 and seven Wuci 2 bud mutation tetraploids, identified and characterised by determining several morphological parameters, flow cytometry and chromosome counting.
Response: Thanks for your positive comments!
- The generation of polyploids by colchicine treatment is a well-known procedure previously applied to many species, includingR. roxburghii. Also, it is doubtful that the specific methods employed here can be directly applied to other taxa since, as rightly pointed out by the authors, the response is highly genotype-dependent. Therefore, the novelty and purely scientific interest of this work is very limited. Nevertheless, the generation of tetraploids in these particular genotypes could have some economic importance considering characteristics of commercial interest of polyploid crop plants, in general, and those of R. roxburghii, in particular, mentioned in the Introduction.
Response: Thanks again for your positive comments!
- 3. Unfortunately, the manuscript has several limitations and drawbacks and will requireextensive modifications and rewriting before it can be considered for publicationin Plants. The most relevant issues to be taken into account when preparing a revised version of the manuscript are detailed below (although the list is not exhaustive):
Response: The manuscript has undergone a substantial number of modifications and rewritings in response to the feedback provided by the reviewers.
- 4. To increase the readers' interest in the reported work, the authors should include inthe Introduction (or maybe in the Discussion) a list of specific commercialapplications of the obtained tetraploids and their possible use in breeding programmes. Some information on this issue is mentioned in the manuscript, but it should be highlighted more clearly.
Response: The information regarding the specific commercial applications of the tetraploids obtained and their potential use in breeding programs is further expounded in Chapter 1 of the manuscript.
- Please follow the journal instructions regarding the organisation and layout of thearticle. For example, Section 2 should be 'Results', not 'Material and Methods'.
Response: The manuscript's organization and layout have been revised in accordance with journal requirements.
- Define all abbreviations at first use in the text
Response: All abbreviations used for the first time in the manuscript have been defined.
- In some cases, the description of the experimental procedures is unclear. Forexample, the authors indicate (section 2.3.1) that they measure several growth and morphological parameters in 'diploid and polyploid strains of R. roxburghii', which is confusing since these measurements are supposed to be used to identify the polyploids. Please, check carefully and rewrite the text where necessary.
Response: The data for analyzing the polyploid phenotype of R. roxburghii f. eseiosa was measured after determining its ploidy. As a result, the results and methods section of the manuscript has been modified.
- 'Chlorophyll meter' (line 173): indicate model, company, city and country.
Response: Information about Chlorophyll meter proposed has been supplemented in Chapter 4.3.3 of the manuscript.
- The statistical analyses are, in general, poorly described. In most cases, the numberof biological replicas used in the experiments is unclear; it is not mentioned in section 2.4 (Data analysis), in the text or in the legends of figures and tables (n = ?). Also, it is nowhere indicated if the numbers in tables and the error bars in figures represent SE or SD.
Response: The description of statistical analysis in the method section of the manuscript has been modified. The numbers in the table in the manuscript and the error bars in the figure represent SD.
- 10. Tables and figures should be understood without reference to the text, including allrelevant information in the table/figure itself or the corresponding legend. The legendsof all tables and figures in the manuscript should be extended to include this information: definition of abbreviations and variables that are not self-evident, definition of all panels in a figure (e.g., panels CK1, CK2, and B to G in Fig. 5), statistical analysis (means ± SE or SD, n = ?), etc.
Response: The manuscript's charts and annotation information have been modified.
11.Some specific comments:
11.1 Table 1: it is unclear; include horizontal lines to separate the different times of treatments and Wuci 1 from Wuci 2.
Response: Table 1 has been modified in the manuscript.
11.2 Fig. 3C is unclear. What do you mean by 'processing diagram'?
Response: The annotation for Fig.3C in the manuscript has been updated to Fig.7, which depicts the plant growth state of Wuci 2 after smear treatment.
11.3 Fig. 4: Y-axis, 'mortality percentage'.
Response: The Y-axis information of Figure 1 in the revised manuscript.
11.4 Fig. 5: The panels showing 'number of stomata', 'DNA content' and 'chromosome number' are too small, and it is not possible to observe the details of the photos. The figure should be modified, or these panels should be shown in independent figures, in any case of larger size.
Response: The figures displaying the stomatal number, DNA content, and chromosome number in the manuscript have been independently presented and edited, and have been assigned to Figures 2 and 3 of the manuscript.
11.5 Do not repeat in the text the numerical values of measured parameters that are included in the tables (e.g., in sections 3.1 or 3.4) since this is redundant and unnecessary.
Response: The measurement parameter values included in the duplicate table in the manuscript results section have been removed.
11.6 The information in the Discussion is interesting, but this section is too long; the text should be shortened.
Response: We've removed some of the discussions in the manuscript.
11.7 Please, use a uniform format for the list of references, following the journal style. For example, the names of the journals are sometimes written in lowercase font and sometimes in capital letters.
Response: The manuscript's references have been revised to comply with the journal's formatting requirements.
- The English language is generally correct. No significant grammar or spelling mistakes have been detected. Nevertheless, the text should be checked to try and improve some sentences that may be somewhat unclear.
Response: The entire manuscript has been carefully examined.

Reviewer 2 Report
Dear editor and authors,
It is a very interesting technical work with great potential of application. I would suggest that it should be published after careful consideration. Details and comments below:
Abstract:
Q1: The authors mention certain characteristics of Wuci samples, such as puncture-free fruit and small sizes. However, it would be important to underline which features are useful for selection programs, as mentioned in the first paragraph of Introtuction.
Q2, When authors mention diploid and monoploid numbers, i.e. 2n=4x=28, “n” and “x” must be in italic format.
Q3: Pehaps is important to mention all the techniques used in these studies, to obtain and check the ploidy levels.
Q4: Please, put keywords in an alphabetical order.
Introduction:
Q5, page 2, line 47: plants do not have hairs, but trichomes. Although this term is commonly used, it would be better to use a botanical term.
Q5, page 2, line 60: replace the word "plants" by "samples", as although polyploidy is more common in plants it occurs in other groups.
Q6, page 2, lines 76-78: The last paragraph of the introduction should address the biological/agronomic problem, or a justification for this scientific investment, but never a short summary of the methodology that was used. I suggest that the authors change this paragraph.
Materials and methods:
Q7, page 3, lines 99-105: in relation to mutagenesis term, I understand that polyploidy is considered a numerical chromosomal mutation involving the entire chromosome set, however, colchicine is not a mutagenic agent from the DNA point of view, since it acts on the GTP's coverage, disturbing the polymerization of tubulins. It might confuse the reader. Is it possible to modify that terminology in the whole manuscript? Perhaps “colchicine solution”, “colchicine treatment”.....
Q8, page 5, lines 177-185: In relation to flow cytommetry assays, it would be good to explain why they did not use a standard, indicate the number of cores analyzed, as well as other important information for the reliability of the experiment.
Q9, page 5, lines 192-193: Please, check the fixative solution, Carnoy’s solution is composed of 60% ethanol, 30% chloroform and 10% glacial acetic acid.
Results:
Q10, page 11, lines 365-367: the chromosome count for tetraploids is completely fragile due to the poor quality of the cells presented in Figure 5. There are many prophases-prometaphases with overlapping chromosomes. While it is apparent that there are more chromosomes in polyploids than in diploids, it is impossible to count them. I suggest replacing all the photos with others with clearly individualized chromosomes.
Discussion:
The authors mention that variations between Wuci 1 and Wuci 2 in the resulting polyploid processes depend on genotypes, but that the information is vague. The fact that colchicine acts on the polymerization of microtubules and the fact that this cytoskeleton is fundamental for several cellular functions, such as movement and positioning of organelles of the endomembrane system, distribution of organelles and chromatids during cell divisions and others, this should be considered in the discussion. The way the discussion is elaborated could induce the reader to understand that colchicine acts as an agent that causes mutations in the genetic material, disregarding the cytoskeleton. For example, observe the following paragraph {In summary, the impregnating seed method produced a low mutation rate in R. roxburghii, while previous studies have demonstrated that the smear method can achieve high mutation rates in various plants. Therefore, this study used the smear method and found it to be suitable for polyploid induction research on R. roxburghii f. eseiosa. Additionally, the mutation rate using the smear method was higher than the mutation rate found with the R. roxburghii seed impregnating method}. For example, if authors change the term "mutations" to "polyploidy", there will be no confusion between the mutation caused in the DNA molecules and numerical chromosome changes due to microtubular damage. In relation to flow cytometry and chromosome observations, I think authors need more accurate informations and imagens, such as graph including pg values of a standard C-value plant and good metaphase plates. Discussion approach themes on polyploid adaptation, stress resistance, changes in plant organs and an increase in yield, accumulation of photosynthetic products and genes regulation, but no experiments have been done on these aspects. I think that the discussion can be shorter and more precise on the progress made in this study.
it is not necessary
Author Response
Dear editor,
Thank you for the positive and constructive comments on our paper entitled “Polyploid induction and identification of Rosa roxburghii f. eseiosa”, Manuscript ID: plants-2393344.
I am very grateful to the reviewers for their suggestions. We have substantially revised our manuscript based on the comments provided by the two reviewers. We have prepared a revised manuscript file with tracked changes and a clean manuscript file. Our responses to the reviewers are outlined below:
Responses to the comments of Reviewer #2
- Abstract:
1.1 Q1: The authors mention certain characteristics of Wuci samples, such as puncture-free fruit and small sizes. However, it would be important to underline which features are useful for selection programs, as mentioned in the first paragraph of Introtuction.
Response: These emphasized contents have been supplemented in the abstract of the manuscript.
1.2 Q2, When authors mention diploid and monoploid numbers, i.e. 2n=4x=28, “n” and“x” must be in italic format.
Response: It has been revised in the manuscript.
1.3 Q3: Pehaps is important to mention all the techniques used in these studies, to obtain and check the ploidy levels.
Response: All techniques used in the study have been outlined in the abstract of the manuscript.
1.4 Q4: Please, put keywords in an alphabetical order.
Response: The keywords have been arranged alphabetically in the manuscript.
- Introduction:
2.1 Q5, page 2, line 47: plants do not have hairs, but trichomes. Although this term is commonly used, it would be better to use a botanical term.
Response: It has been modified in Chapter 1(lines 45-46) of the manuscript.
2.2 Q5, page 2, line 60: replace the word "plants" by "samples", as although polyploidy is more common in plants it occurs in other groups.
Response: In chapter 1 ( line 56 ) of the manuscript, the word ' plant ' was replaced with ' sample '.
2.3 Q6, page 2, lines 76-78: The last paragraph of the introduction should address the biological/agronomic problem, or a justification for this scientific investment, but never a short summary of the methodology that was used. I suggest that the authors change this paragraph.
Response: The final paragraph of Chapter 1 in the manuscript has been modified.
3 Materials and methods:
3.1 Q7, page 3, lines 99-105: in relation to mutagenesis term, I understand that polyploidy is considered a numerical chromosomal mutation involving the entire chromosome set, however, colchicine is not a mutagenic agent from the DNA point of view, since it acts on the GTP's coverage, disturbing the polymerization of tubulins. It might confuse the reader. Is it possible to modify that terminology in the whole manuscript? Perhaps “colchicine solution”, “colchicine treatment”.....
Response: The term "mutagen" throughout the manuscript has been revised to refer to colchicine induction and colchicine treatment.
3.3 Q8, page 5, lines 177-185: In relation to flow cytommetry assays, it would be good to explain why they did not use a standard, indicate the number of cores analyzed, as well as other important information for the reliability of the experiment.
Response: As long as the ratio of lysate to dye in the flow cytometry is 1:4, it is sufficient. So based on the sample characteristics of R. roxburghii f. eseiosa, we can adjust the 1600 μL system to an 800 μL system. We have supplemented information about the flow cytometer in Figure 2 of the manuscript.
3.4 Q9, page 5, lines 192-193: Please, check the fixative solution, Carnoy’s solution is composed of 60% ethanol, 30% chloroform and 10% glacial acetic acid.
Response: The configuration of Carnoy 's fixative in Chapter 4.3.2 of the examined manuscript is : ethanol : glacial acetic acid, 3 : 1. Reference to literature such as Colchicine- and trifluralin-mediated polyploidization of Rosa multiflora Thunb. var. inermis and Rosa roxburghii f. normalis、In vitro induction of tetraploids in Vitis × Muscadinia hybrids、In vitro tetraploid plants regeneration from leaf explants of multiple genotypes in Populus etc.
4 Results:
4.1 Q10, page 11, lines 365-367: the chromosome count for tetraploids is completely fragile due to the poor quality of the cells presented in Figure 5. There are many prophases-prometaphases with overlapping chromosomes. While it is apparent that there are more chromosomes in polyploids than in diploids, it is impossible to count them. I suggest replacing all the photos with others with clearly individualized chromosomes.
Response: The images depicting chromosome numbers in the manuscript have been replaced with clearly defined images of individual chromosomes.
5 Discussion:
The authors mention that variations between Wuci 1 and Wuci 2 in the resulting polyploid processes depend on genotypes, but that the information is vague. The fact that colchicine acts on the polymerization of microtubules and the fact that this cytoskeleton is fundamental for several cellular functions, such as movement and positioning of organelles of the endomembrane system, distribution of organelles and chromatids during cell divisions and others, this should be considered in the discussion. The way the discussion is elaborated could induce the reader to understand that colchicine acts as an agent that causes mutations in the genetic material, disregarding the cytoskeleton. For example, observe the following paragraph {In summary, the impregnating seed method produced a low mutation rate in R. roxburghii, while previous studies have demonstrated that the smear method can achieve high mutation rates in various plants. Therefore, this study used the smear method and found it to be suitable for polyploid induction research on R. roxburghii f. eseiosa. Additionally, the mutation rate using the smear method was higher than the mutation rate found with the R. roxburghii seed impregnating method}. For example, if authors change the term "mutations" to "polyploidy", there will be no confusion between the mutation caused in the DNA molecules and numerical chromosome changes due to microtubular damage. In relation to flow cytometry and chromosome observations, I think authors need more accurate informations and imagens, such as graph including pg values of a standard C-value plant and good metaphase plates. Discussion approach themes on polyploid adaptation, stress resistance, changes in plant organs and an increase in yield, accumulation of photosynthetic products and genes regulation, but no experiments have been done on these aspects. I think that the discussion can be shorter and more precise on the progress made in this study.
Response: In the discussion section of the manuscript, it has been clarified that the difference between Wuci 1 and Wuci 2 depends on the genotype. Moreover, discussions on the mechanism of colchicine action have been added, and the word 'mutation' has been replaced with 'polyploid'. Additionally, clearer pictures have been used to replace flow cytometry and chromosome pictures. However, discussions related to polyploid adaptability, stress resistance, changes in plant organs and yield increase, accumulation of photosynthetic products, and gene regulation have been deleted.
Kind regards,
Huijing Wu, Lanlan Jiang, Jin'e Li, Min Lu and Hua-Ming An
Guizhou University

Round 2
Reviewer 1 Report
The manuscript has been substantially improved and could be accepted for publication once some minor formal/format issues, which still remain, are corrected.
- even though the authors provide more data on the statistical analyses in the Materials and Methods section, this information must also be included in the legends of figures and tables to make them understandable without reference to the text (e.g. "values are means +/- SD, n = 3")
- please check the numbering of the subsections in the text and the reference to the panels in the legend of Fig. 3.
The English language is correct. No major issues detected
Author Response
- The manuscript has been substantially improved and could be accepted for publication once some minor formal/format issues, which still remain, are corrected.
Response: Thank you very much for your feedback. The manuscript has been revised.
- even though the authors provide more data on the statistical analyses in the Materials and Methods section, this information must also be included in the legends of figures and tables to make them understandable without reference to the text (e.g. "values are means +/- SD, n = 3")
Response: The annotations to the charts in the manuscript provide further information on the statistical analysis.
- please check the numbering of the subsections in the text and the reference to the panels in the legend of Fig. 3.
Response: I have checked the numbering of each section in the manuscript and checked the citation position of Figure 3 in the manuscript.
Reviewer 2 Report
Please check for formatting and typos throughout the manuscript. Please improve the focus of the bottom images in Figure 2. For the chromosome images, it is possible to improve the quality if the background is made a little lighter and the chromosomes a little more contrasted.
Minor language tweaks are needed, especially to shorten long sentences and make text clearer.
Author Response
Responses to the comments of Reviewer
- Please check for formatting and typos throughout the manuscript. Please improve the focus of the bottom images in Figure 2. For the chromosome images, it is possible to improve the quality if the background is made a little lighter and the chromosomes a little more contrasted.
Response: I have checked the format and corrected typos in the entire manuscript. In addition, I have modified Figure 2 in order to improve the display of chromosome numbers.
- Minor language tweaks are needed, especially to shorten long sentences and make text clearer.
Response: The grammar of some lengthy sentences in the manuscript has been improved to enhance the clarity of the text.